# Disease Outcome and Brain Metabolomics of Cyclophilin-D Knockout Mice in Sepsis

**DOI:** 10.3390/ijms23020961

**Published:** 2022-01-16

**Authors:** Takayuki Kobayashi, Hiroyuki Uchino, Eskil Elmér, Yukihiko Ogihara, Hidetoshi Fujita, Shusuke Sekine, Yusuke Ishida, Iwao Saiki, Shoichiro Shibata, Aya Kawachi

**Affiliations:** 1Department of Anesthesiology, Tokyo Medical University, Tokyo 160-0023, Japan; h-uchi@tokyo-med.ac.jp (H.U.); yogihara@tokyo-med.ac.jp (Y.O.); shus@tokyo-med.ac.jp (S.S.); yishida@tokyo-med.ac.jp (Y.I.); iwao9876@gmail.com (I.S.); shbt1118@gmail.com (S.S.); a-kwc311@tokyo-med.ac.jp (A.K.); 2Mitochondrial Medicine, Department of Clinical Sciences, Lund University, 221 84 Lund, Sweden; eskil.elmer@med.lu.se; 3Department of Biomedical Engineering, Osaka Institute of Technology, Osaka 535-8585, Japan; hidetoshi.fujita@oit.ac.jp

**Keywords:** encephalopathy, mitochondria, oxidative stress, glutathione, cyclophilin D

## Abstract

Sepsis-associated encephalopathy (SAE) is a diffuse brain dysfunction resulting from a systemic inflammatory response to infection, but the mechanism remains unclear. The mitochondrial permeability transition pore (MPTP) could play a central role in the neuronal dysfunction, induction of apoptosis, and cell death in SAE. The mitochondrial isomerase cyclophilin D (CypD) is known to control the sensitivity of MPTP induction. We, therefore, established a cecal ligation and puncture (CLP) model, which is the gold standard in sepsis research, using CypD knockout (CypD KO) mice, and analyzed the disease phenotype and the possible molecular mechanism of SAE through metabolomic analyses of brain tissue. A comparison of adult, male wild-type, and CypD KO mice demonstrated statistically significant differences in body temperature, mortality, and histological changes. In the metabolomic analysis, the main finding was the maintenance of reduced glutathione (GSH) levels and the reduced glutathione/oxidized glutathione (GSH/GSSG) ratio in the KO animals following CLP. In conclusion, we demonstrate that CypD is implicated in the pathogenesis of SAE, possibly related to the inhibition of MPTP induction and, as a consequence, the decreased production of ROS and other free radicals, thereby protecting mitochondrial and cellular function.

## 1. Introduction

In this study, the analysis of sepsis-associated encephalopathy (SAE) was performed from a new perspective using cyclophilin D-knockout (CypD KO) mice. Sepsis is characterized by a robust systemic inflammatory response induced by infection. Despite improvements in antibiotic therapies and critical care management, sepsis is among the most common reasons for patient admission to the intensive care unit, and is associated with a high mortality rate [1,2,3]. Sepsis-induced multiple organ failure is clinically characterized by cerebral, pulmonary, cardiovascular, renal, and gastrointestinal dysfunction. SAE, or sepsis-associated delirium, is common among patients in intensive care units and is associated with an increased morbidity and mortality [4,5,6]. The diagnosis of SAE and its effects on mortality and morbidity have not been analyzed in detail to date. This is probably in part owing to the lack of precise, well-established, clinical, and biological markers for its diagnosis, and it is, therefore, frequently not recognized [7,8,9]. SAE is defined as any brain dysfunction associated with septic illness and a diffuse cerebral dysfunction induced by sepsis without a direct central nervous system infection [10]. Clinical symptoms of SAE include delirium, a fluctuating mental status, changes in attention, and disorganized thinking, and SAE of variable severity has been found to occur in 9% to 71% of septic patients [4,5]. The pathogenesis of SAE is complex, and its pathophysiology remains unclear to date, but is suggested to include mitochondrial and endothelial dysfunction, neurotransmission disturbances, the disruption of calcium homeostasis in brain tissue, and blood flow dysfunction [11,12] pathways in the CLP model of sepsis.

## 2. Results

### 2.1. Sepsis-Induced Hypothermia and Mortality

After CLP treatment, the WT group displayed lower body temperatures than the KO group (*n* = 10 per group). At 0 h, the mean body temperatures of the WT and KO mice were not significantly different, at 36.2 ± 0.2 °C and 36.5 ± 0.3 °C (*p* = 0.058), respectively. However, 18 h after the CLP treatment, the body temperatures of the WT group decreased to 29.8 ± 1.3 °C, whereas those of the KO group were maintained at 32.0 ± 1.7 °C, resulting in a statistically significant difference between the two groups (*p* = 0.008). 

Survival rates were compared between the two groups (*n* = 10 per group), and a significant difference was observed (*p* = 0.004). In the WT group, half of the mice died within 30 h after CLP treatment, and all died within 46 h. In the KO group, more than half of the animals survived for 70 h or longer (Figure 1 and Figure 2).

### 2.2. Effects of Cyclophilin D KO on Neuronal Cell Death

Celestine blue and acid fuchsin staining was performed to analyze neuronal cell death in both the parietal cortex and hippocampus (including the CA1, CA2, and CA3 regions) 18 h after the CLP treatment (Figure 3A,B). Neuronal damage (densely stained cells with a high nuclear/cytoplasmic ratio, cellular swelling, as well as neuronal vacuolization) in the cerebral cortex and the hippocampus was counted and displayed as the percentage of damaged neurons (of total number of cells) (Figure 3C,D). At 0 h after the CLP treatment, there was no significant difference between the two groups regarding neither the cerebral cortex nor the hippocampus. At 18 h, neuronal cell death was significantly more severe in both the cerebral cortex and hippocampus in the WT group as compared to the KO group (cortex: *p* < 0.001, hippocampus: *p* = 0.004). Specifically, in the cerebral cortex, necrotic cells in the WT group were increased from 3.79% to 63.8% from 0 h to 18 h, while those in the KO group changed from 3.45% to 10.1% from 0 h to 18 h. In the hippocampus, necrotic cells in the WT group increased from 4.29% to 10.5% from 0 h to 18 h, while those in the KO group changed from 3.53% to 4.65% from 0 h to 18 h. In both groups, signs of cerebral edema-like changes were observed following CLP, and the brain was damaged, but when comparing the number of necrotic cells, the WT group was more seriously damaged (Figure 3).

### 2.3. Metabolomics of Brain Tissue

A metabolomic analysis was performed on mouse brains by CE-TOFMS. The detected candidate compounds were classified into glycolysis/gluconeogenesis, the pentose phosphate pathway, the citric acid cycle, the urea cycle, purine/pyrimidine metabolism, coenzyme metabolism, and various amino acid metabolism pathways, and the pathways of the metabolites were mapped. Significant differences were obtained between the KO and WT groups for a number of metabolites (Figure 4A, green meaning significantly underexpressed and red overexpressed, respectively) (see also Appendix A), but the changes in the reduced glutathione (GSH), and the precursor of phosphatidylcholine and potential neuroprotective intermediary CDP-Choline (citicoline) were the most relevant to the analysis of neuronal dysfunction in sepsis. The KO group maintained a higher GSH value than the WT group at all timepoints (0, 6, and 18 h), which reached statistical significance at 18 h (*p* = 0.022) (Figure 4C). In addition, when comparing the sham group (without CLP) with the WT group with CLP, GSH was found to be significantly reduced 18 h after CLP treatment (*p* = 0.029, data not shown). A similar profile was also found for the GSH/GSSG ratio, which was significantly higher in the KO group at 18 h (Figure 4D). Citicoline was also maintained at a high level in the KO group at 6 and 18 h after CLP (significant at 6 h, *p* = 0.043) (Figure 4C). Furthermore, the WT group after CLP treatment was compared with the sham group, and the citicoline levels of the WT group were found to be significantly lower than those of the sham group at 6 h after CLP treatment (*p* = 0.032, data not shown). When calculating the ratio of the choline break-down product betaine with citicoline (Figure 4D), the KO group maintained a higher ratio at both 6h (*p* = 0.003) and 18 h (*p* = 0.009) following CLP as compared to the WT group.

## 3. Discussion

In this study, the maintenance of body temperature, a significant improvement in survival rate, and a significant reduction in brain damage were observed in the CypD KO group compared with the wild-type group, following the induction of sepsis.

In addition, the metabolomic analysis demonstrated that the levels of GSH and citicoline were maintained at high levels in the KO group. GSH is a thiol-containing compound and exerts antioxidant effects by reacting with ROS to become oxidized GSH (glutathione disulfide, GSSG). Citicoline (CDP-Choline) is another endogenous compound known to relieve oxidative stress, suppress mitochondrial-mediated apoptosis pathways, and exert neuroprotective effects by crossing the blood–brain barrier (BBB) and reaching the central nervous system [13]. By knocking out CypD, we created an environment that made it easier to avoid the induction of mitochondrial permeability transition. Maintaining the mitochondrial function and ATP-producing ability likely protected antioxidant molecules such as GSH and citicoline, which could have influenced the outcome of sepsis. Our results supported the notion that mitochondrial protection may reduce brain damage and cell death. Temperature maintenance, survival rate, and brain cell damage displayed more favorable results in the KO group. Further, GSH and citicoline levels were found to be significantly higher in the KO group in the metabolomic analysis of brain tissue. The original idea for this study came from a previous study on ischemic neuronal cell death from our group. First, the immunosuppressive agent cyclosporin A (CsA), a calcineurin inhibitor, was shown to have brain-protective effects against ischemic brain injury and traumatic brain injury [14,15]. CsA was known to bind to CypD, which is a component of the mitochondrial permeability transition pore (MPTP), and inhibited the MPTP opening [16,17,18,19]. Following the induction of MPTP, the subsequent release of cytochrome C and the activation of caspases were thought to induce mitochondrial dysfunction and cell death [20,21]. However, as CsA has a low BBB permeability, we focused on its target CypD in this study. CypD, which is encoded by the Ppif gene, belongs to a family of enzymes called the peptidyl prolyl isomerases, exists in the mitochondrial matrix, and is considered to be involved in MPTP opening as a component of the multiple protein complex [22,23], forming the mitochondrial transition pore. In brain tissue, we analyzed the differences in metabolites between CLP model mice and WT mice to assess the role of CypD. The most interesting result was that GSH and citicoline levels remained significantly high in KO mice compared with WT mice in the metabolomic analysis. GSH is synthesized as a tripeptide of glutamic acid, cysteine, and glycine. It is actively biosynthesized from the component amino acids by ATP-dependent ligase. In addition, it exists in a reduced form (GSH) and an oxidized form (GSSG), and has an antioxidant effect that eliminates reactive oxygen species (such as O_2_, H_2_O_2_, and OH) through their reduction [24,25,26]. Examples of enzymes that scavenge reactive oxygen species and, hence, have antioxidant effects include SOD (superoxide dismutase), catalase, and GSH. A study on alcoholic liver injury has demonstrated that alcohol reduces intramitochondrial GSH, promotes caspase activation, and apoptosis induction owing to a decrease in the mitochondrial membrane potential and an increase in membrane permeability [27,28,29]. Furthermore, the overproduction of inflammatory cytokines and ROS occurs during liver damage, causing further oxidative damage to the mitochondrial membrane, followed by cell death and cell necrosis [30,31,32]. In the present study, the difference in GSH maintenance in the brain (and other tissues) may have promoted the significant difference in the outcomes between the two groups. This was further confirmed by the maintained GSH/GSSG ratio in the KO group, suggesting a protected antioxidant effect.

Citicoline is an essential intermediate in the biosynthetic pathway of cell membrane structural phospholipids, particularly phosphatidylcholine [13,33,34]. Once absorbed, citicoline is widely distributed throughout the body, crosses the BBB, reaches the central nervous system, and is incorporated into phospholipid membranes, such as microsomes. Citicoline activates the biosynthesis of structural phospholipids in nerve cell membranes, enhances brain metabolism, acts on various neurotransmitters, and exerts neuroprotective effects in hypoxia and ischemic conditions [34,35,36]. Citicoline has also been reported to improve learning and memory capacity in animal models of aging [37,38]. In addition, citicoline has been shown to prevent mitochondrial dysfunction and to improve cerebral edema in various experimental models. It has also been reported to inhibit the apoptotic pathways associated with cerebral ischemia and specific neurodegenerative models (associated with caspases 3 and 9) [39,40]. More interestingly, in experiments using human neuroblastoma cells and the forebrain ischemia model in mice, citicoline has been shown to prevent the decrease in reduced GSH, and reduces caspase 3 activation, which causes apoptosis [41,42]. Further, in this study, the citicoline/betaine ratio was clearly lowered by sepsis. The acetylcholine metabolic pathway is related to the Kennedy pathway and exposure to oxidative stress can affect both betaine and citicoline and possibly the acetylcholine metabolism. Similar to the GSH/GSSG ratio, the CypD knockouts maintained a higher citicoline/betaine ratio following sepsis. 

In this study, we demonstrated that the inhibition of the mitochondrial permeability transition via the genetic knockout of the mitochondrial matrix isomerase cyclophilin D positively affected the disease phenotype and mortality in the CLP model of sepsis in adult male mice. The direct causal relationship between CypD knockout and the improved outcome following the induction of sepsis is not yet clear. However, decreasing the probability of MPTP opening could protect cellular energy levels, decrease signals of apoptosis (via cytochrome c release), as well as decrease the generation of free radicals from mitochondria. This could possibly lead to the preservation of antioxidative defenses as demonstrated in the present study. Further studies are warranted to clarify the role of the mitochondrial permeability transition, oxidative stress, and metabolic pathways in the CLP model of sepsis.

## 4. Materials and Methods

### 4.1. Sepsis Model 

The study design was approved by the Animal Care and Use Committee of Tokyo Medical University (study approval number; H31-0087), and all procedures involving animals were performed in accordance with the institutional and national guidelines for animal experimentation. Adult male wild-type (WT) mice and mice null for Ppif on the same background (Ppif-/- mice) were obtained from The Jackson Laboratory (Bar Harbor, ME, USA). Mice had free access to food and water, and were housed in a temperature-controlled room at 24 to 28 °C on a 12 h light–dark cycle. To induce sepsis, male C57/BL6 mice at 10 to 16 weeks of age (*n* = 53) were anesthetized with 4% to 5% sevoflurane, and a middle abdominal incision was performed along the ventral surface of the abdomen to expose the cecum. Before perforation, the feces were gently relocated toward the distal cecum. The cecum was ligated at 5 mm from the distal end and, subsequently, punctured once with a 21-gauge needle, allowing exposure of feces (cecal ligation and puncture; CLP). A small droplet of feces was then gently squeezed through both sides of the puncture. The cecum was returned to the peritoneal cavity with careful attention so that feces did not contaminate the margins of the abdominal and skin wound. The muscle and skin incision were closed with 3-0 black silk. Mice receiving sham surgery underwent the same procedure, except that the cecum was neither ligated nor punctured (sham group; Sham). At each experimental timepoint, the mice were sacrificed by decapitation, and brain tissue samples were collected.

### 4.2. Measurement of Body Temperature and Survival Rate

Body temperature of the mice was measured by rectal thermometry, which is a common method of measuring body temperature in rodents. Mice were hand-restrained and placed on a cage lid. The tail was then lifted, and a probe was gently inserted into the rectum to a fixed depth for measurement. For measuring survival rate, we set the endpoint at 72h after the CLP to measure survival rate and avoid prolongation of disease period more than necessary, and we checked the life and death of the mice every hour.

### 4.3. Histological Analysis

Brain samples were obtained 18 h after the CLP procedure or sham-operation (*n* = 5 per group). Samples were fixed by immersion in 10% buffered formaldehyde overnight, embedded in paraffin, and cut into 6 mm thick coronal sections. The slides were stained with a combination of celestine blue and acid fuchsin or hematoxylin and eosin for histopathological analysis. Quantification of brain damage was performed in a blinder manner by direct visual counting of acidophilic neurons at a magnification of 400× using an optical microscope [15]. Necrotic cells (both the nuclei and cytoplasm) were stained intensely acidophilic, and the shape of the nucleus varied from triangular to oval. Small fragments of acidophilic dendritic processes, even if undergoing phagocytosis, were not counted. Neuronal damage in the hippocampus and in the parietal cortex was counted at the level of the bregma minus 2.1 mm, and presented as the percentage number of damaged neurons of the total number of cells.

### 4.4. Metabolomic Analysis

Analysis of 230 metabolites was performed using capillary electrophoresis time of flight mass spectrometry (CE-TOFMS), and the results of septic mice and sham mice were compared. Metabolites that were quantitatively measured included those involved in glycolysis, the pentose phosphate pathway, the tricarboxylic acid (TCA) cycle, and the urea cycle, as well as purine, pyrimidine, GSH, nicotinamide, and amino acid metabolism. Hippocampal samples were obtained at 0 h, 6 h, and 18 h after the CLP procedure, and at 0 h, 6 h, and 18 h after the sham operation (*n* = 3 per group). Approximately 20 mg of frozen brain tissue was placed in 1500 mL of 50% acetonitrile/Milli-Q water containing internal standards (H3304-1002, Human Metabolome Technologies Inc., Tsuruoka, Japan) at 0 °C. The hippocampus was homogenized three times at 1500 rpm for 120 s using a tissue homogenizer (BMS-M10N21, BMS Co., Ltd., Tokyo, Japan) and then the homogenate was centrifuged at 2300× *g* (4 °C) for 5 min. Subsequently, 800 μL of the upper aqueous layer was centrifugally filtered through a Millipore 5-kDa cutoff filter at 9100× *g* and 4 °C for 120 min to remove proteins. The filtrate was centrifugally concentrated and resuspended in 50 μL of Milli-Q water for CE-TOFMS analysis. Metabolomic measurements were carried out by Human Metabolome Technologies Inc.

### 4.5. Statistical Analysis

Data are expressed as means ± SD. Statistical analysis was performed using Excel and SPSS (IBM^®^ SPSS^®^ Statistics 27). Comparison of three or more datasets was performed by one-way analysis of variance (one-way ANOVA). The Student *t*-test was used for comparison between two groups. The Kaplan–Meier method and log-rank test were used to analyze survival rates. Differences were considered to be statistically significant when the *p*-value was less than 0.05.

## Figures and Tables

**Figure 1 ijms-23-00961-f001:**
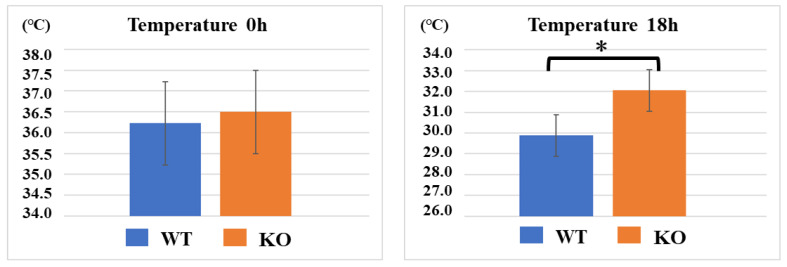
Sepsis-induced hypothermia. At 0 h, the mean body temperatures of the WT and KO mice were not significantly different, at 36.2 °C and 36.5 °C, respectively. However, 18 h after the CLP treatment, the body temperatures of the WT group decreased to 29.9 °C, whereas those of the KO group were maintained at 32.0 °C. Data are expressed as means ± SD, * *p* < 0.008, *n* = 10.

**Figure 2 ijms-23-00961-f002:**
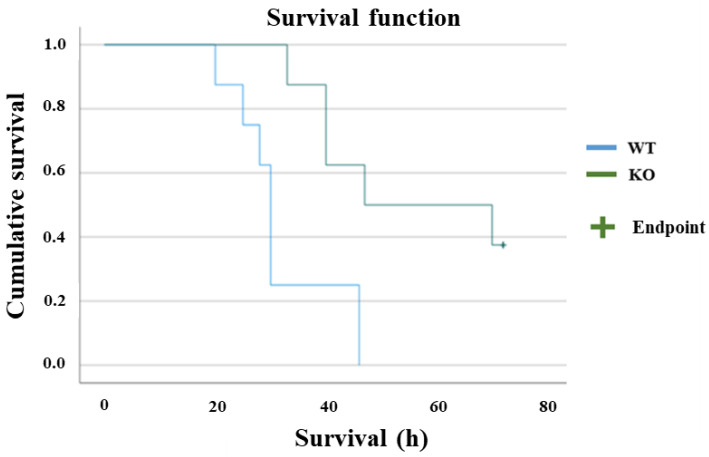
Survival plot. The survival rates were compared between the two groups (*n* = 10 per group) with log-rank test, and a significant difference was observed (*p* = 0.004). In the WT group, half of the mice died within 30 h after CLP treatment, and all died within 46 h. In the KO group, more than half of the animals survived for 70 h or longer.

**Figure 3 ijms-23-00961-f003:**
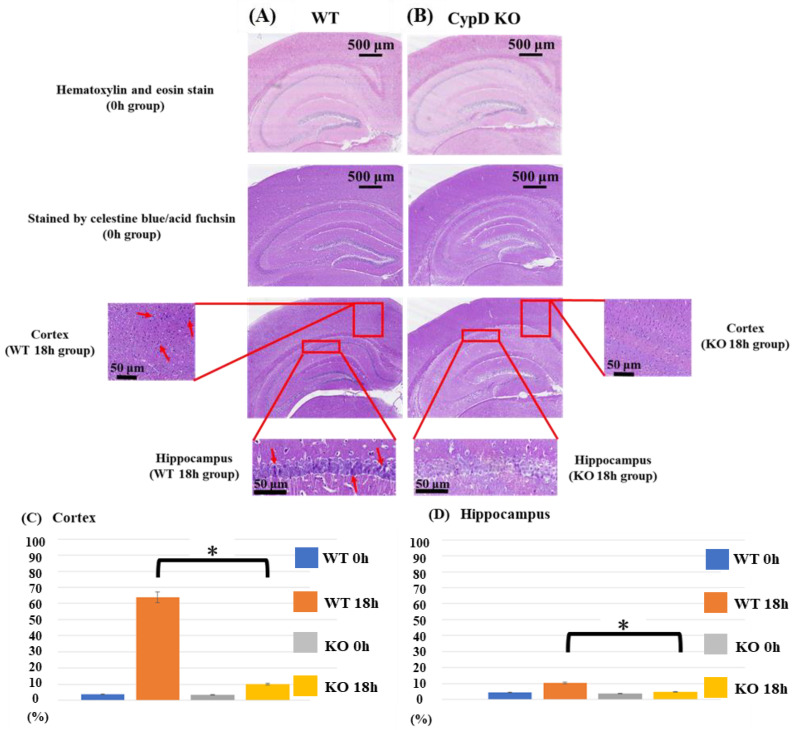
Histological analysis. Images from acid-fuchsin and celestine blue-stained sections represent neuronal necrosis of cortex and hippocampal areas from WT (sham-operated control (**A**) and from CypD KO mice (**B**)). Neuronal damage in the cerebral cortex and the hippocampus was counted in a blinded manner and presented as the percentage number of damaged neurons (of the total number of cells). Tissues were harvested 18 h after surgery. Arrows indicate densely stained cells with high nuclear/cytoplasmic ratio, cellular swelling, and neuronal vacuolization. At 0 h after CLP treatment, there was no significant difference between the two groups regarding neither cerebral cortex nor hippocampus. At 18 h, neuronal cell death was significantly more severe in both the cerebral cortex and hippocampus in the WT group as compared to the KO group (**C**,**D**). Data are expressed as means ± SD, * *p* < 0.05, *n* = 5.

**Figure 4 ijms-23-00961-f004:**
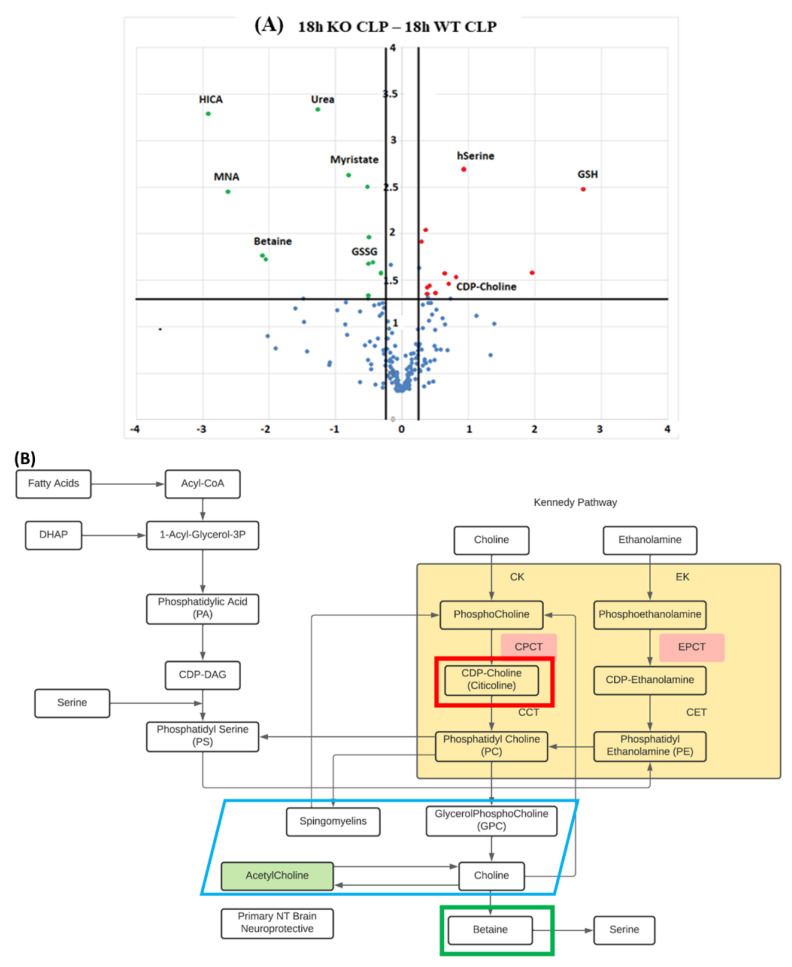
Metabolomics of brain tissue following CLP. (**A**) Volcano plot for differential brain tissue metabolite expression 18 h following induction of sepsis in CypD KO vs. WT mice. Scattered points represent individual metabolites: the *x*-axis is the Log 2 (fold change) for the ratio KO-CLP-18 h vs. WT-CLP-18 h, whereas the *y*-axis is the Log (*p*-value), where *p*-value is the probability that a metabolite has statistical significance in its differential expression. Red dots are, thus, metabolites significantly over-expressed in KO-CLP-18 h, and green dots are metabolites significantly underexpressed in KO-CLP-18 h vs. WT-CLP-18 h. (**B**) CDP-Choline/citicoline and betaine were intermediaries related to the Kennedy pathway of phosphatidylcholine synthesis. The KO group maintained a higher brain GSH value than the WT group at all timepoints (0, 6, and 18 h), which was statistically significant at 18 h. (**C**) Citicoline was also maintained at a high level in the KO group at 6 and 18 h after CLP. (**D**) Focusing on the redox effect and calculating the ratio of GSH to GSSG or the ratio of citicoline to betaine, both ratios were maintained at a higher level in the KO group. Data are expressed as means ± SD, * *p* < 0.05, *n* = 3.

## Data Availability

Data is contained within the article or Supplementary Material.

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
