# Peer review of "Disease Outcome and Brain Metabolomics of Cyclophilin-D Knockout Mice in Sepsis"

_ijms, 2022, doi:10.3390/ijms23020961_

Round 1
Reviewer 1 Report
In this paper Kobayashi, et al, examined the effect of genetic deletion of CyD, an essential protein component of the mitochondrial transition pore complex, on rectal temperature, mortality and metabolomic alterations in mice following fecal soiling of the peritoneal cavity by cecal ligation/puncture (CLP). CyD deletion blunted hypothermia, decreased mortality, decreased hippocampal neuronal loss and maintained hippocampal GSH/GSSG ratios and citicholine levels. Their findings support the involvement of CyD and thus the mitochondrial transition pore involvement in oxidative stress and likely in septic encephalopathy.
This is important research that is straightforward and likely to be of interest in sepsis research and even in care of septic patients (would reduction of oxidative stress improve outcomes?) I feel the paper would benefit from the following changes, which would require mostly minor modifications:
- GSH is traditionally referred to as "reduced glutathione" and GSSG as "oxidized glutathione". It would be helpful to the reader to use these terms consistently throughout the manuscript.
- lines 191-196. The discussion of liver effects detracts from the Discussion and could be eliminated.
- "pore" should be stated as "mitochondrial transition pore" (as the authors know, there are many "pores" throughout the body.
- Are wild-type (WT) mice the proper control? Is there instead a genetic control for the CyD KO mice? At the minimum the authors should discuss this point.
- Fig 3B. the high power microscopic views should be even higher power to show the neuronal degeneration better.
- What about other (non-hippocampal) brain regions? esp cortical regions?
- Do the authors plan on studying any tissues other than brain? Because the CyD KO is body-wide, the authors have a nice model to examine other high-energy, poorly replicating tissues such as heart and skeletal muscle.
Reviewer 2 Report
International Journal of Molecular Sciences
Special Issue: Mitochondria-Targeted Approaches in Health and Disease 2.0
IJMS-1506189: “The mitochondrial permeability transition in sepsis – disease outcome and brain metabolomics of cyclophilin-D knockout mice“
COMMENTS TO THE AUTHORS
Dear Authors,
Please find enclosed the comments for the above-mentioned manuscript.
A SUMMARY OF THE CONTENT
The authors stated that the study intends to analyze the disease phenotype, and possible onset mechanism of SAE (sepsis-associated encephalopathy) by metabolomic analyses of brain tissue. The authors explained rationale to use CypD (cyclophilin-D) knockout mice since CypD is known to control the sensitivity of MPTP (mitochondrial permeability transition pore) implicated in acute neurodegeneration. The cecal ligation and puncture model (CLP) as the gold standard in sepsis research was applied. Results showed that CypD-KO-mice demonstrated statistically significant differences in body temperature, mortality, and histological changes. The main findinge of the metabolomic analysis were maintenance of GSH (glutathione levels) and the GSH/GSSG ratio in the CypD-KO-mice animals following. The authors concluded that CypD is implicated in the pathogenesis of SAE, possibly related to inhibition of MPTP induction and as a consequence decreased production of ROS (reactive oxygen species) and other free radicals thereby protecting mitochondrial and cellular function.
THE OVERALL OPINION OF THE MANUSCRIPT
The manuscript is within the scope of the journal and describes the original and important topic. The topic is of interest to the readers of IJMS. The main limitations are: the title and the conclusions are not supported by the results, the results are not precisely described, the sex of animal was not specified although it is important and the sex-different response was not shown, the pioneered work was some part of the text introduction.
Accordingly, the major and substantial revision involving the new experiments related to MPTP (mitochondrial permeability transition pore) and sex-different-response results is required for future consideration.
(1) TITLE
The title is not supported by data. Please perform new experiments to keep the title and show the status and/or the role of MPTP.
(2) ABSTRACT
2.1. Please focus on the aim, the methodology and the results, rather than introduction. Namely more than 8 (44.44%) rows out of 18 describe background.
2.2. Please replace “the molecular mechanism” with “possible molecular mechanism” since the precise mechanism was not described.
(3) INTRODUCTION
3.1. Please be more precise in describing.
3.2. Please cite pioneered work related to the subject as well as recent advanced in the field (this year publications).
(4) METHODS
4.1. Please provide the number of the males and females.
4.2. Please perform experiments to show results on males and females and please use more homogenous group in term of aging since there are differences in hormonal status between 10 weeks and 14 weeks animal. It is very well known that hormonal status highly affects metabolism and functionality.
4.3. Please provide new molecular results related to MPTP (mitochondrial permeability transition pore) if you want to keep the title.
4.4. Please provide intra- and inter- assay coefficients for all assays.
(5) RESULTS
5.1. Please be more clear and precise in the description of the results.
5.2. Please provide the new results from the new experiments related to MPTP (mitochondrial permeability transition pore) and sex-different-response.
(6) DISCUSSION
6.1. Please discuss the new results from the new experiments related to MPTP (mitochondrial permeability transition pore) and sex-different-response.
6.2. Please cite pioneered work related to the subject as well as recent advanced in the field (this year publications).
(7) FIGURES
Please provide the new figures showing the new results from the new experiments related to MPTP (mitochondrial permeability transition pore) and sex-different-response.
(8) GENERAL
Please use official abbreviations.
Good luck and all the best :)
Reviewer 3 Report
This study demonstrates that inhibition of mitochondrial permeability transition through cyclophilin D genetic knockout attenuates disease phenotype and mortality in the CLP model of sepsis. Based on the metabolomic analysis, the authors hypothesize that the effect may be related to the demonstrated conservation of antioxidant defenses, given the maintenance of reduced glutathione levels. However, the analysis is still preliminary and further studies are needed to define the direct causal correlation between CypD knockout and improved outcome after sepsis induction.
Round 2
Reviewer 2 Report
International Journal of Molecular Sciences
Special Issue: Mitochondria-Targeted Approaches in Health and Disease 2.0
IJMS-1506189R1: “Disease outcome and brain metabolomics of cyclophilin-D 3 knockout mice in sepsis“
COMMENTS TO THE AUTHORS
Dear Authors,
Please find enclosed the comments for the above-mentioned manuscript.
THE OVERALL OPINION OF THE MANUSCRIPT
THE OVERALL OPINION OF THE REVISED MANUSCRIPT
The manuscript is within the scope of the journal and describes the original important topic. The topic is of interest to the readers of IJMS, but does not fit to the special issue. The authors clarified some issues, but not important. The main limitations are: the title and the conclusions are not supported by the results, the results are not precisely described.
Accordingly, the substantial revision including the discussion related to sex-different response is required for future consideration.
Please find enclosed some of the suggestions in the comments to the authors.
(1) TITLE
The title is not supported by data. Please clearly state that your study is performed only in male and only in adult male.
(2) ABSTRACT
2.1. Please focus on the aim, the methodology and the results, rather than introduction.
(3) INTRODUCTION
3.1. Please be more precise in describing.
(4) METHODS
4.1. Please provide intra- and inter- assay coefficients for all assays.
(5) RESULTS
5.1. Please be more clear and precise in the description of the results.
(6) DISCUSSION
6.1. Please discuss the sex-different response since it is very well known that hormonal status highly affects metabolism and functionality.
Good luck and all the best :)
Reviewer 3 Report
The reviewer's main criticism remains regarding the comment that the analysis is still preliminary and further studies are needed to define the direct causal correlation between CypD knockout and improved outcome after sepsis induction. In the revised version of the work, no new experimental information was added to investigate or at least discuss the mechanism involved. The sentence reported in discussion that emphasizes that "further studies are needed to define the causal relationship between CypD knockout and improvement in outcome after sepsis" is not sufficient and was also present in the previous version.
Round 3
Reviewer 3 Report
he manuscript has been sufficiently improved to warrant publication in IJMS